# Vertical AI-driven Scientific Discovery

## Abstract

Automating scientific discovery has been a grand goal of Artificial Intelligence (AI) and will bring tremendous societal impact if it succeeds. Despite exciting progress, most endeavor in learning scientific equations from experiment data focuses on the *horizontal* discovery paths, i.e., they directly search for the best equation in the full hypothesis space. Horizontal paths are challenging because of the associated exponentially large search space. Our work explores an alternative *vertical* path, which builds scientific equations in an incremental way, starting from one that models data in *control variable experiments* in which most variables are held as constants. It then extends expressions learned in previous generations via adding new independent variables, using new control variable experiments in which these variables are allowed to vary. This vertical path was motivated by human scientific discovery processes. Experimentally, we demonstrate that such vertical discovery paths expedite symbolic regression. It also improves learning physics models describing nano-structure evolution in computational materials science.

## 1 Introduction

Automating scientific discovery has been a grand goal of Artificial Intelligence (AI) dating back its founders (Herbert Simon et. al. [13, 11, 30]) but remains a holy grail. The underlying societal impact is immense because of its multiplier effect. Indeed, much effort has been made, especially in symbolic equation regression, including search-based methods [12, 14], genetic programming [26, 29, 24, 5], reinforcement learning [21, 25, 18, 21], deep function approximation [17, 2, 23, 22, 16, 31, 3, 7, 1], integrated systems [28, 10, 9, 15], or simply yet effectively, collecting big datasets [15, 8]. Most endeavor focuses on *horizontal* discovery paths, i.e., they directly search for the best equation in the full hypothesis space involving all independent variables (red path in Figure 1). The horizontal search can be challenging because of the exponentially large space. After the conventional wisdom of training with larger models and more data has been stretched to its extremity (e.g., GPT-4), what is the next paradigm-changing idea?

Interestingly, the *vertical* paths have been largely overlooked in AI. To discover the ideal gas law $pV = nRT$, scientists first held $n$ (gas amount) and $T$ (temperature) as constants and find $p$ (pressure) is inversely proportional to $V$ (volume). They then studied the relationship between $pV$ and $n, T$. This led to a vertical discovery path (green path in Figure 1). The first few steps of a vertical path can be significantly cheaper than the horizontal path, because the searches are in reduced spaces involving a small number of

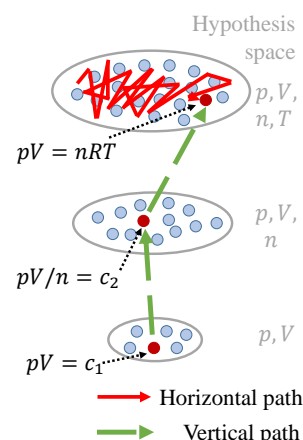

Figure 1: Vertical paths further scale up AI-driven scientific discovery.

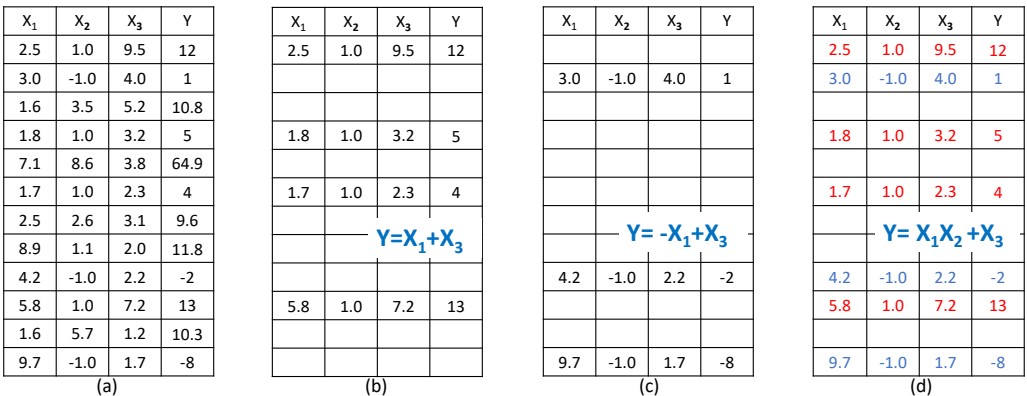

Figure 2: Motivating example to demonstrate vertical scientific discovery. **(a)** A challenging symbolic regression task. It is difficult to read out the equation $y = f(x_1, x_2, x_3)$ which connects the dependent variable $y$ with the independent variables $x_1, x_2, x_3$. **(b)** When we focus on studying the relationship of $x_1$, $x_3$ and $y$ while holding $x_2$ at 1, a simple equation $y = x_1 + x_3$ can be discovered. **(c)** $y = -x_1 + x_3$ can be discovered when we hold $x_2$ at -1. **(d)** Combining (b) and (c), a good candidate equation is $y = x_1 x_2 + x_3$, which turns out to be the ground-truth equation.

independent variables. As a result, vertical discovery has the potential to supercharge state-of-the-art approaches in modeling complex scientific phenomena with more interlocking contributing factors or processes than what current approaches can handle.

This paper demonstrates the power of vertical scientific discovery, in which automated reasoning (e.g., mathematical programming, constraint satisfaction, reinforcement learning, etc) acts as robot scientists to guide the learning process (i.e., pointing out the directions of the green path in Figure 1).

Our first example is in symbolic regression, where the task is to discover symbolic expressions describing experiment data. State-of-the-art approaches in this domain are limited to learning simple expressions. Regressing expressions involving many independent variables still remain out of reach. Motivated by the control variable experiments widely utilized in science, in a recently published paper [6] we propose **C**ontrol **V**ariable **G**enetic **P**rogramming (CVGP) for symbolic regression over many independent variables. CVGP expedites symbolic expression discovery via customized experiment design, rather than learning from a fixed dataset collected a priori. CVGP starts by fitting simple expressions involving a small set of independent variables using genetic programming, under controlled experiments where other variables are held as constants. It then extends expressions learned in previous generations by adding new independent variables, using new control variable experiments in which these variables are allowed to vary. Experimentally, CVGP outperforms several baselines in learning symbolic expressions involving multiple independent variables.

Our second example is in materials science. Our approach was motivated by tracking and learning the phase-field models describing nano-scale crystalline defect evolution in materials. In a preliminary study, we showed vertical discovery schedules improve the learning of phase-field models for dendritic solidification. In the vertical schedule, first the learning is concentrated on a subset of model parameters. This is done by feeding the model with designed training data in which remaining parameters do not affect the spatial and temporal dynamics. After this phase, the learning is expanded to all parameters. We demonstrate that the machine learning model is able to discover the ground-truth phase-field model following this vertical schedule, but cannot following the normal schedule (see the Figure in Section 4).

## 2 A Motivating Example

Discovering scientific laws automatically from experiment data has been a grand goal of Artificial Intelligence (AI). Its success will greatly accelerate the pace of scientific discovery. Recently, exciting progress [26, 29, 4, 21, 18, 21, 24, 5] has been made in this domain, especially taking advantages of the progress in deep neural networks. Consistent strides for higher *throughput* (less time and data

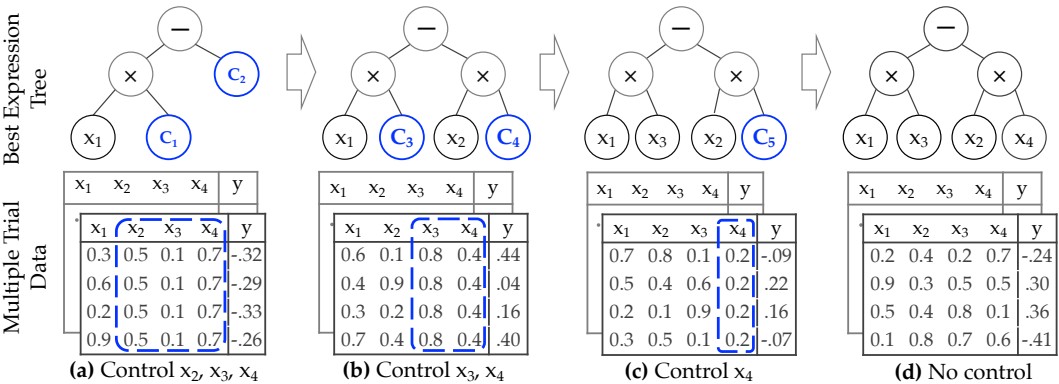

Figure 3: Running example of CVGP. **(a)** Initially, a reduced-form equation $\phi' = C_1 x_1 - C_2$ is found via fitting control variable data in which $x_2, x_3, x_4$ are held as constants and only $x_1$ is allowed to vary. **(b)** This equation is expanded to $C_3 x_1 - C_4 x_2$ in the second stage via fitting the data in which only $x_3, x_4$ are held as constants. **(c,d)** This process continues until the ground-truth equation $\phi = x_1 x_3 - x_2 x_4$ is found. The data generated for control variable experiment trials in each stage are shown at the bottom.

required to identify an equation) and better *quality* (equations better fit the data) have been the main drivers in this domain. We notice that almost all prior work follows the horizontal discovery path, which is also the standard machine learning pipeline – first collecting a dataset, then learning the full model, finally evaluating its performance on a separate, yet still fixed test set. Nevertheless, the ***vertical discovery path***, which is heavily utilized by human scientists, is almost forgotten in AI-driven scientific discovery. When studying a complex process involving many interacting subprocesses, scientists always try to isolate each individual process and study their effects separately, via carefully designed control variable experiments. They also use this tool to challenge competing models.

***Vertical paths increase the throughput of scientific discovery***. Let us verify this assertion from a small human experiment. Figure 2 (a) depicts a symbolic regression task where one needs to find a symbolic expression $y = f(x)$ which best maps the input $x$ to the output $y$. The author ran this experiment in front of hundreds of undergraduate, graduate students, and a few faculty members. Nobody was able to discover the correct equation given the data in (a). However, when the author controlled the value of $x_2$ in (b) and (c), a majority of the audience were able to identify the equations in both cases. A little bit of additional thinking combining these two equations yields the ground-truth equation in (d). Clearly, control variable experiments in (b) and (c) helped the audience navigate the regression task. This controlled experiment depicts the essence of vertical scientific discovery.

## 3 Symbolic Regression via Control Variable Genetic Programming

Our recently proposed **C**ontrol **V**ariable **G**enetic **P**rogramming (CVGP) [6] implements the vertical scientific discovery process using Genetic Programming (GP) for symbolic regression over many independent variables. The key insight of CVGP is to learn from *a customized set of control variable experiments*; in other words, the experiment data collection adapts to the learning process. This is in contrast to the current learning paradigm of most symbolic regression approaches, where they learn from a fixed dataset collected a priori.

In CVGP, first, we hold all independent variables except for one as constants and learn an expression that maps the single variable to the dependent variable using GP. GP maintains a pool of candidate expressions and improves the fitness of these equations via mating, mutating, and selection over several generations. Mapping the dependence of one independent variable is easy. Hence GP can usually recover the ground-truth reduced-form equation. Then, CVGP frees one independent variable at a time. In each iteration, GP is used to modify the equations learned in previous generations to incorporate the new independent variable, via mating, mutating, and selection. Such a procedure repeats until all the independent variables have been incorporated into the symbolic expression. See figure 3 for the high-level idea of algorithm execution. Theoretically, in the original paper we

Table 1: Median (50%) and 75%-quantile Normalized Mean Squared Error (NMSE) values of the symbolic expressions found by all the algorithms on several *noisy* benchmark datasets (Gaussian noise with zero mean and standard deviation 0.1 is added). Our CVGP finds symbolic expressions with the smallest NMSEs.

| Dataset configs | CVGP (ours) 50% | CVGP (ours) 75% | GP 50% | GP 75% | DSR 50% | DSR 75% | PQT 50% | PQT 75% | VPG 50% | VPG 75% | GPMeld 50% | GPMeld 75% |
|---|---|---|---|---|---|---|---|---|---|---|---|---|
| (4,4,6) | **0.036** | **0.088** | 0.038 | 0.108 | 1.163 | 3.714 | 1.016 | 1.122 | 1.087 | 1.275 | 1.058 | 1.374 |
| (5,5,5) | 0.076 | 0.126 | **0.075** | **0.102** | 1.028 | 2.270 | 1.983 | 4.637 | 1.075 | 2.811 | 1.479 | 2.855 |
| (5,5,8) | **0.061** | **0.118** | 0.121 | 0.186 | 1.004 | 1.013 | 1.005 | 1.006 | 1.002 | 1.009 | 1.108 | 2.399 |
| (6,6,8) | **0.098** | **0.144** | 0.104 | 0.167 | 1.006 | 1.027 | 1.006 | 1.020 | 1.009 | 1.066 | 1.035 | 2.671 |
| (6,6,10) | **0.055** | **0.097** | 0.074 | 0.132 | 1.003 | 1.009 | 1.005 | 1.008 | 1.004 | 1.015 | 1.021 | 1.126 |
| **(a) Datasets containing operators $\{\sin, \cos, \mathrm{inv}, +, -, \times\}$.** | | | | | | | | | | | | |
| (3,2,2) | **0.098** | **0.165** | 0.108 | 0.425 | 0.350 | 0.713 | 0.351 | 1.831 | 0.439 | 0.581 | 0.102 | 0.597 |
| (4,4,6) | **0.078** | **0.121** | 0.120 | 0.305 | 7.056 | 16.321 | 5.093 | 19.429 | 2.458 | 13.762 | 2.225 | 3.754 |
| (5,5,5) | **0.067** | **0.230** | 0.091 | 0.313 | 32.45 | 234.31 | 36.797 | 229.529 | 14.435 | 46.191 | 28.440 | 421.63 |
| (5,5,8) | **0.113** | **0.207** | 0.119 | 0.388 | 195.22 | 573.33 | 449.83 | 565.69 | 206.06 | 629.41 | 363.79 | 666.57 |
| (6,6,8) | **0.170** | **0.481** | 0.186 | 0.727 | 1.752 | 3.824 | 4.887 | 15.248 | 2.396 | 7.051 | 1.478 | 6.271 |
| (6,6,10) | **0.161** | **0.251** | 0.312 | 0.342 | 11.678 | 26.941 | 5.667 | 24.042 | 7.398 | 25.156 | 11.513 | 28.439 |
| **(b) Datasets containing operators $\{\sin, \cos, +, -, \times\}$.** | | | | | | | | | | | | |
| (3,2,2) | 0.049 | **0.113** | **0.023** | 0.166 | 0.663 | 2.773 | 1.002 | 1.992 | 0.969 | 1.310 | 0.413 | 2.510 |
| (4,4,6) | **0.141** | **0.220** | 0.238 | 0.662 | 1.031 | 1.051 | 1.297 | 1.463 | 1.051 | 1.774 | 1.093 | 1.769 |
| (5,5,5) | **0.157** | 0.438 | 0.195 | **0.337** | 1.098 | 3.617 | 1.018 | 5.296 | 1.012 | 1.27 | 1.036 | 3.617 |
| (5,5,8) | **0.122** | **0.153** | 0.166 | 0.186 | 1.009 | 1.103 | 1.017 | 1.429 | 1.007 | 1.132 | 1.07 | 2.904 |
| (6,6,8) | **0.209** | **0.590** | **0.209** | 0.646 | 1.003 | 1.153 | 1.047 | 1.134 | 1.059 | 1.302 | 1.029 | 3.365 |
| (6,6,10) | 0.139 | 0.232 | **0.073** | **0.159** | 1.654 | 3.408 | 1.027 | 1.069 | 1.009 | 1.654 | 1.445 | 2.106 |
| **(c) Datasets containing operators $\{\sin, \cos, \mathrm{inv}, +, -, \times\}$.** | | | | | | | | | | | | |

show CVGP as an incremental builder can reduce the exponential-sized search space for candidate expressions into a polynomial one when fitting a class of symbolic expressions. Experimentally, we show CVGP outperforms a number of state-of-the-art approaches on symbolic regression over multiple independent variables (see Table 1).

# 4 Vertical Scientific Discovery in Modeling Nano-structure Evolution in Materials Science

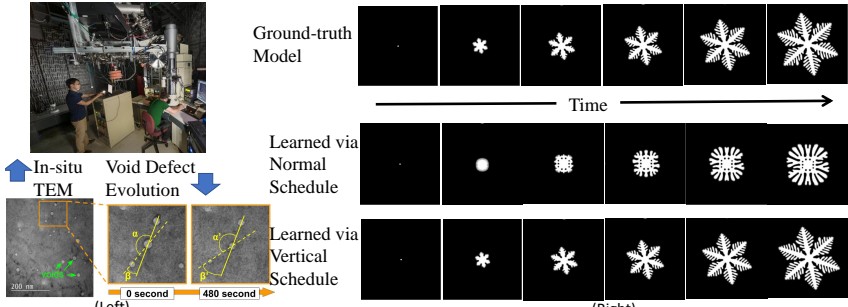

We intend to apply the idea of vertical scientific discovery in learning nano-scale defect evolution for material under extreme conditions. Nano-scale crystalline defects can appear in different forms in these materials. Extreme environments of heat and irradiation can cause these defects to evolve in size and position. As shown in the left panel of the figure above, void shaped defects are captured by transmission electron microscope (TEM) cameras during in-situ radiation experiments. These defects appear in round shapes, and drift in position as demonstrated by the change of angles to respectively, as time progresses. They also change size. These changes can affect the physical and mechanical properties of the material. For this reason, characterizing these defects is essential in designing new materials that can resist adverse environments. Collaborating with materials scientists, we have been analyzing terabytes of in-situ TEM videos of this type and have already made scientific discoveries [27, 32, 20, 19].

As a preliminary study, vertical discovery schedules are used to improve the learning of phase-field models for dendritic solidification. In the vertical schedule, first the learning is concentrated on a subset of model parameters. This is done by feeding the model with designed training data in which the remaining parameters do not affect the dynamics of the PDEs. After this phase, the learning is expanded to all parameters. The right panel of the figure above demonstrate that learning via the vertical schedule is able to identify the correct phase-field model while normal schedules cannot.

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
