# OpenReview forum: "Vertical AI-driven Scientific Discovery"
_NeurIPS.cc/2023/Workshop/AI4Science — NeurIPS2023-AI4Science Poster_

### Official Review · Reviewer_hGSh · 2023-10-16
**The authors have provided valuable insights and evidence of symbolic AI from the perspective of vertical driven methods that make this paper potentially publishable.**

**Rating:** 6
**Confidence:** 3

**Review:**

This article discusses the concept of vertical scientific discovery, which involves building scientific equations incrementally by starting with simple models that only consider a few variables and then gradually extending them to include more variables. The paper contributes to the existing literature in a meaningful way as this approach is inspired by how human scientists conduct experiments by isolating variables and studying their effects separately.

The first example focuses on symbolic regression, where the goal is to discover symbolic expressions that describe experimental data. The authors propose a method called Control Variable Genetic Programming (CVGP), which uses genetic programming to learn equations in an incremental manner. CVGP starts by fitting simple expressions involving a small set of variables using genetic programming and then expands these expressions by incorporating new variables. Experimental results show that CVGP outperforms several baselines in learning symbolic expressions involving multiple independent variables. More details should be covered to enhance the quality of the work.

The second example is in the field of materials science, specifically in modeling nano-scale crystalline defect evolution. The authors demonstrate how a vertical discovery schedule, which starts with a subset of model parameters and gradually expands to include all parameters, improves the learning of phase-field models for dendritic solidification. I am a little surprised about the readouts of this experiment. However, it will be more persuasive if more examples are added.

Overall, the article highlights the potential benefits of vertical scientific discovery in accelerating AI-driven scientific discovery and suggests that it can enhance symbolic regression and improve the learning of complex scientific phenomena.

---

### Official Review · Reviewer_xBE8 · 2023-10-18
**A great work on general AI for Science**

**Rating:** 8
**Confidence:** 3

**Review:**

This article introduces the concept of **vertical scientific discovery**, a methodology involving incremental construction of scientific equations. It commences with simple models that encompass only a limited number of variables and progressively broadens them to encompass additional variables. This approach mirrors how human scientists conduct experiments, where variables are isolated and their individual impacts studied, making it a valuable addition to existing scientific literature.

The first case study concentrates on symbolic regression, a pursuit to uncover symbolic expressions that elucidate experimental data. The authors introduce Control Variable Genetic Programming (CVGP), a method employing genetic programming to learn equations step by step. CVGP initiates the process by fitting straightforward expressions involving a small variable set using genetic programming. It then systematically expands these expressions by integrating new variables. Empirical evidence indicates that CVGP surpasses several baseline methods when it comes to acquiring symbolic expressions that involve multiple independent variables. Nevertheless, there is room for a more comprehensive exploration of these results to enhance the overall quality of the research.

The second example lies within the domain of materials science, specifically concerning the modeling of nanoscale crystalline defect evolution. The authors illustrate how a vertical discovery schedule, which begins with a subset of model parameters and gradually encompasses all parameters, enhances the mastery of phase-field models for dendritic solidification. While the findings from this experiment are somewhat unexpected, a more convincing argument could be made by including additional examples.

In summary, this paper is well-written and easy to follow and underscores the potential advantages of vertical scientific discovery in expediting AI-driven scientific exploration. It suggests this approach can enhance symbolic regression and elevate the understanding of intricate scientific phenomena, although further instances would strengthen this assertion.

Also, I suggest the author extend these two examples and add more examples if possible.

I recommend the acceptance.